# Beyond Euler: An Explainable Machine Learning Framework for Predicting and Interpreting Buckling Instabilities in Non-Ideal Materials

## Abstract

Predicting structural failure is a fundamental objective in materials science and mechanical engineering. Euler's classical formula, the standard for predicting the buckling instability of slender columns for over 250 years, assumes idealized material properties that can lead to unreliable predictions and potentially catastrophic failures in critical infrastructure. This study proposes a solution by introducing a novel framework that synergizes machine learning and modern explainability techniques to model complex physical systems. We used pasta as a model non-ideal material for a comprehensive experimental analysis and a dataset from 147 controlled buckling experiments on four distinct pasta gauges. We then developed a physics-informed XGBoost model, incorporating both raw geometric measurements and a composite feature derived from Euler's formula ($G = d^4/L^2$), and subsequently evaluated the model's performance using a 5-fold cross-validation scheme. The model demonstrated an outstanding predictive power, achieving an average coefficient of determination ($R^2$) of 0.97 and a Root Mean Squared Error (RMSE) of 0.14 N. We also examined the model's internal decision-making process by employing SHAP (SHapley Additive exPlanations). The analysis confirmed the primary importance of the theoretically-derived feature but also revealed that the model learned to use raw geometric data as crucial correction factors. This study presents a powerful proof of concept for using interpretable machine learning to achieve not only predictive accuracy but also gain deeper physical insights into complex, non-ideal systems. The framework presented here has broad implications for advancing our understanding and design capabilities in materials science, engineering, and advanced manufacturing.

## 1 Introduction

### 1.1 The Engineering Imperative of Stability Analysis

Civil and mechanical engineering have long focused on predicting the response of materials to applied forces (Simitses and Hodges, 2006; Bazant and Cedolin, 2009). The reliability of any structure depends on its ability to remain stable under its prescribed operational loads (ASCE/SEI, 2022; CEN, 2005). Accurate predictions of instabilities are essential to prevent catastrophic collapses, which can incur severe economic and human costs (ASCE, 2021). Among the various modes of structural failure, the phenomenon of buckling stands out as particularly insidious (Thompson, 2016). Unlike a material slowly yielding under tension, a column undergoing buckling can transition from a state of stable equilibrium to total failure with little to no warning (Brush and Almroth, 1975). This is not merely a theoretical concern; it is a real-world failure mode seen in critical infrastructure, from columns in buildings and bridges to sub-sea pipelines (DNV, 2021) and railway tracks that can buckle under the compressive forces of thermal expansion (FRA, 2011). Accurately predicting the onset of buckling proves itself to be an engineering imperative of the highest order (CEN, 2005).

## 1.2 The Classical Framework: Euler's Buckling Theory

Leonhard Euler achieved the first successful mathematical description of column buckling in 1744. His work established a foundational pillar of structural mechanics and provided an equation that remains central to engineering education and practice (Timoshenko and Gere, 1961). Euler's critical load formula, shown in Eq. (1), defines the maximum axial compressive load an ideal column can sustain before it becomes unstable (Timoshenko and Gere, 1961).

$$P_{cr} = \frac{\pi^2 EI}{(KL)^2} \tag{1}$$

To fully appreciate this equation, it is necessary to understand its constituent terms: $P_{cr}$ (the critical load), E (Young's Modulus of Elasticity), I (Area Moment of Inertia), L (Effective Length), and K (Effective Length Factor).

## 1.3 The Gap Between Ideal Theory and Physical Reality

The elegance and enduring power of Euler's formula lie in its foundation of mathematical idealism; however, this is also its primary limitation in real-world scenarios. A significant and well-documented "reality gap" exists because no physical object perfectly satisfies the formula's core assumptions (Timoshenko and Gere, 1961). The key sources of this discrepancy are:

- **Material Heterogeneity and Anisotropy:** Euler's formula assumes the material is homogeneous and isotropic. Many real-world materials, however, do not meet this ideal. For example, natural materials like wood have grain and growth rings (Guinea et al., 2004), and modern engineered materials like 3D-printed polymers exhibit anisotropic properties due to their layered construction (Harmon et al., 2021). Pasta, as an extruded product, exhibits brittle failure modes inconsistent with the above ideal materials (Carpentieri et al., 2024; Arbelo et al., 2014).
- **Geometric Imperfections:** The formula assumes a perfectly straight column. All real objects exhibit minute variations that create eccentricities where the applied load is not perfectly aligned with the column's central axis, inducing bending moments that the ideal theory does not account for (Brush and Almroth, 1975; Vemareddy et al., 2022).
- **Difficulty in Parameter Estimation:** The formula's accuracy depends critically on the value of Young's Modulus, E. For pasta, this value has been investigated using various methods, yielding a range of results and highlighting the difficulty in characterization (Vargas-Calderón et al., 2019; Hou, 2010; Gladden et al., 2005).
- **Non-Linear Material Behavior:** The theory assumes linear elasticity. Many materials, including pasta, exhibit non-linear and brittle behavior, failing suddenly without significant prior elastic deformation (Carpentieri et al., 2024; Gladden et al., 2005).

These limitations mean that any attempt to use Euler's formula to precisely predict the failure of a real-world object is fraught with uncertainty. Building on prior studies in the field, our study focuses on this gap, using pasta as an accessible and illustrative example of a non-ideal material.

## 1.4 A New Paradigm: Data-Driven Modeling with Explainable AI

To bridge this reality gap, we turn to a new paradigm: supervised machine learning. The application of machine learning to predict column buckling is an active area of research, with recent studies successfully using models like artificial neural networks for braced columns (Chadha et al., 2021) and corrugated steel girders (Kalyoncuoglu et al., 2024). Instead of beginning with a theoretical formula, a data-driven approach inverts the process, learning the complex and underlying associations directly from experimental observations. We hypothesize that a machine learning model, when presented with measurable geometric features, can learn a highly accurate mapping to the experimentally observed buckling load.

High predictive accuracy alone, however, is not sufficient for scientific inquiry. An AI model that acts as an impenetrable black box offers little new physical insight. This brings us to the core novelty of this study: the integration of Explainable AI (XAI), a growing field of study in engineering and

computer science (Love et al., 2022). By using state-of-the-art XAI techniques, specifically SHAP (SHapley Additive exPlanations) (Lundberg and Lee, 2017), we aim to "open the black box" and understand how the model arrives at its predictions.

### 1.5 STATEMENT OF CONTRIBUTIONS

This study makes the following novel contributions:

- It provides a rigorous, quantitative demonstration of the limitations of Euler's classical buckling formula when applied to a common, non-ideal material.

- It develops and validates a high-performance XGBoost Machine Learning model that predicts the critical buckling load with exceptionally high accuracy ($R^2$=0.97), outperforming the theoretical model.

- It successfully integrates a physics-informed feature derived from classical theory into a machine learning pipeline, demonstrating a powerful synergy between the two approaches.

- Most importantly, it employs a state-of-the-art XAI technique (SHAP) to provide a deep, mechanistic interpretation of the model's decision-making process, revealing the subtle, non-linear correction factors that the model learns to apply to the classical framework.

The remainder of this paper is structured as follows: Section 2 details the experimental methods and computational framework. Section 3 presents the model's predictive performance and the results of the explainability analysis. Finally, Section 4 discusses the implications of our findings and concludes the paper.

## 2 MATERIALS AND METHODS

### 2.1 EXPERIMENTAL APPARATUS

We designed the experimental setup for replicability and precision based on a standard engineering curriculum. The experimental apparatus, as depicted in Figure 1, consisted of a digital scale (with a precision of ±0.01 g), standard masking tape, a ruler, and a digital caliper (±0.01 mm). This minimalist setup allows for the direct measurement of the critical buckling force.

### 2.2 DATA COLLECTION PROTOCOL

We followed a meticulous protocol to ensure data consistency.

- **Sample Preparation:** 147 Individual strands of pasta were selected and visually inspected for major defects.

- **Geometric Measurement:** For each selected strand, the total length (L) was measured once. The diameter (d) was measured at three distinct points and averaged.

- **Buckling Test:** A single pasta strand was placed vertically on the surface of the digital scale, which was subsequently zeroed. We used a small piece of tape to secure the bottom of the strand to the scale's surface to prevent slipping. Using an index finger, we applied a slow, steady, and vertically aligned compressive force to the top of the pasta strand.

- **Load Measurement:** The critical buckling point was identified as the maximum mass reading on the scale at the precise moment the pasta strand underwent sudden lateral deformation.

- **Calculation:** The critical load ($P_{cr}$) in Newtons was calculated by multiplying the measured mass (in kg) by the standard acceleration due to gravity (9.81 m/s²).

### 2.3 DATASET CHARACTERISTICS

The final dataset comprised 147 distinct observations from four pasta types: Spaghetti, Angel Hair, Thin Spaghetti, and Vermicelli; These four types span a broad diameter range (approximately

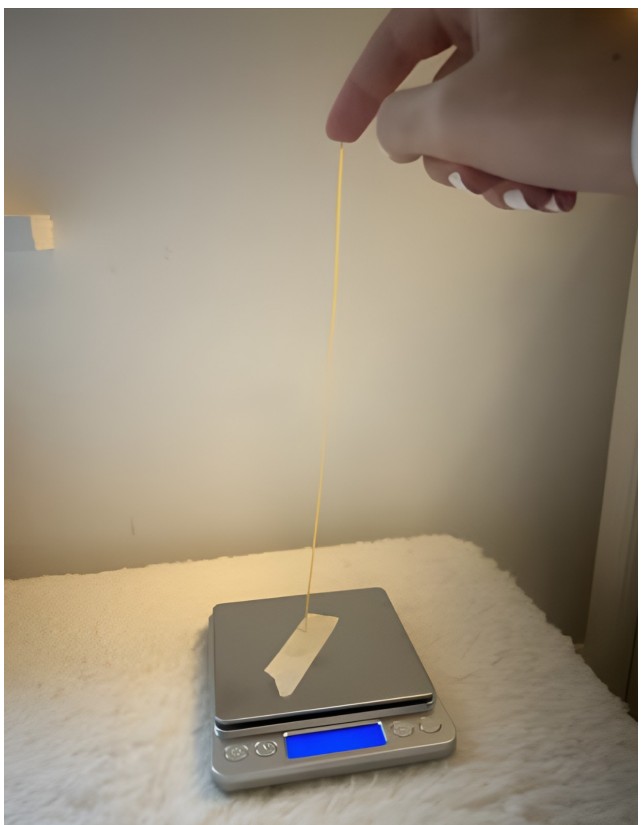

Figure 1: The experimental setup, showing a pasta strand positioned vertically on a digital scale, ready for compressive force to be applied by finger.

1.2–1.7 mm) and introduce controlled manufacturing heterogeneity, thereby improving model generalization and enabling explainability analyses. Statistical analysis revealed a well-distributed set of experimental conditions and outcomes, summarized in Table 1.

Table 1: Descriptive Statistics of Final Combined Experimental Data (N=147).

| Variable | Mean | Std. Dev. | Min | Max |
|---|---|---|---|---|
| Length (cm) | 14.0 | 4.1 | 8.0 | 20.0 |
| Diameter (mm) | 1.48 | 0.21 | 1.2 | 1.7 |
| Load $P_{cr}$ (N) | 0.81 | 0.82 | 0.09 | 3.22 |

### 2.4 COMPUTATIONAL METHODOLOGY

*Feature Engineering:* Raw features included length (m), diameter (m), and a categorical variable `pasta_type` (one-hot encoded). A physics-informed feature, G, was engineered from Euler's formula, $G = d^4/L^2$. This approach aligns with the emerging paradigm of Theory-Guided Data Science (Karpatne et al., 2017).

*Gradient Boosting Machine (XGBoost):* We selected the Extreme Gradient Boosting (XGBoost) algorithm (Chen and Guestrin, 2016), implemented in Python using the Scikit-learn library (Pedregosa et al., 2011).

*Model Training and Validation:* To ensure robust performance, we employed a 5-fold cross-validation scheme.

*Explainable AI (XAI) with SHAP:* To interpret the model, we used SHAP (Lundberg and Lee, 2017), a game theory-based approach that calculates the contribution of each feature to each individual prediction.

## 3 RESULTS

### 3.1 PREDICTIVE ACCURACY OF THE GRADIENT BOOSTING MODEL

The 5-fold cross-validation process provided a robust estimate of the model's performance on unseen data. The averaged metrics were exceptional:

- Coefficient of Determination ($R^2$): 0.97
- Root Mean Squared Error (RMSE): 0.14 N

An $R^2$ value of 0.97 indicates that our model can explain 97% of the variance in the experimental buckling loads. The RMSE of 0.14 N signifies a very low average prediction error, given that measured loads ranged up to 3.22 N. This high accuracy is visualized in the Predicted vs. Actual plot (Figure 2).

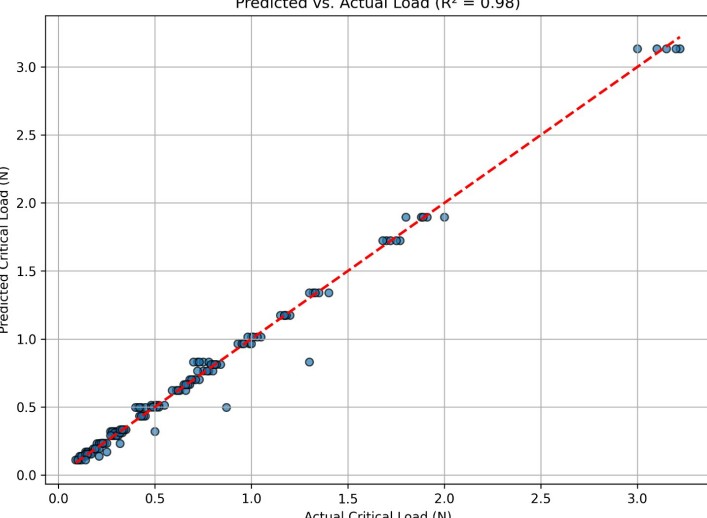

Figure 2: Predicted vs. Actual critical buckling load (N). The tight clustering along the diagonal line ($R^2$=0.97) demonstrates exceptional accuracy from the XGBoost Model vs. the Experimental Data.

### 3.2 MODEL INTERPRETATION WITH EXPLAINABLE AI

A SHAP summary plot (Figure 3) provides a comprehensive overview of global feature importance. The analysis reveals that the physics-informed feature, G, is the most influential predictor. However, the raw 'diameter' and 'length' features remain highly important, suggesting the model is learning subtle correction factors beyond the scope of the classical formula.

### 3.3 WALKTHROUGH OF A SINGLE PREDICTION

To illustrate how the model makes a prediction, we can examine a single sample. Consider a spaghetti strand with a length of 12 cm and a diameter of 1.7 mm. The experimentally measured buckling load for this strand was 1.32 N. SHAP analysis reveals how each feature contributes a specific value to push the prediction away from the baseline average (0.81 N).

- The baseline average prediction is 0.81 N.

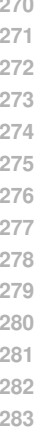
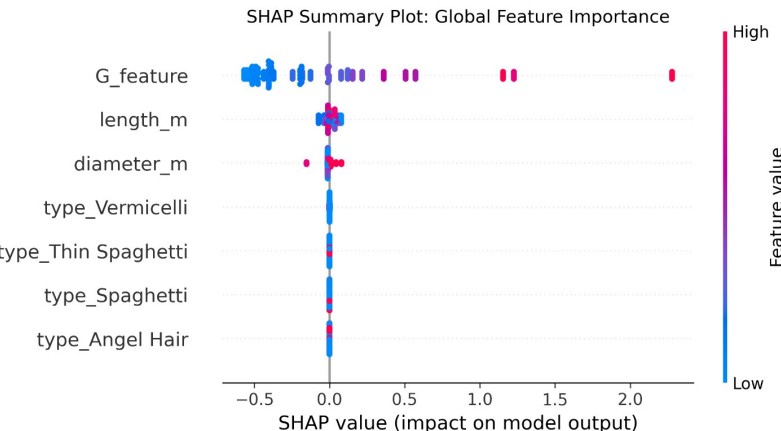

Figure 3: SHAP summary plot showing global feature importance.

- The high value of the physics-informed feature (G) provides a SHAP value of +0.45 N.
- The relatively high diameter provides another push, with a SHAP value of +0.12 N.
- The moderate length has a small negative impact, with a SHAP value of -0.03 N.

Summing these contributions (0.81 + 0.45 + 0.12 - 0.03), the model arrives at a final predicted load of 1.35 N, very close to the experimental value of 1.32 N.

### 3.4 COMPARATIVE ANALYSIS

To place the performance of the XGBoost model in context, we compared it against a Random Forest model and the classical Euler formula, for which we estimated an average Young's Modulus (E) of approximately 2.9 GPa from the dataset. The results are summarized in Table 2.

Table 2: Performance Comparison of Predictive Models

| Model Type | Model Name | R² | RMSE (N) |
|---|---|---|---|
| Theoretical | Euler's Formula | 0.81 | 0.38 |
| Machine Learning | Random Forest | 0.96 | 0.16 |
| **Machine Learning** | **XGBoost (Ours)** | **0.97** | **0.14** |

Random Forest (RF) averages independently grown trees (lower variance, higher bias), whereas gradient boosting fits residuals sequentially and captures interactions/curvature—hence the lower RMSE and higher R² (Chen and Guestrin, 2016).

## 4 DISCUSSION

The results validate our central hypothesis that a hybrid data-driven approach can bridge the gap between idealized theory and experimental reality. The outperformance of the XGBoost model demonstrates the limitations of a purely theoretical approach for non-ideal materials. The success of the physics-informed feature, G, shows that classical theory remains invaluable, providing a strong foundation upon which the machine learning model can build. This synergy aligns with the principles of Theory-Guided Data Science (Karpatne et al., 2017).

The most novel contribution is using XAI for scientific insight. The independent importance of "diameter" and "length" is the key finding. It implies the true relationships are more complex than simple power laws. For example, the non-linear importance of 'diameter' may suggest the model is

learning about the relationship between a column's thickness and its resistance to localized crushing at the contact points, an effect not present in Euler's pure bending theory. Similarly, the independent importance of 'length' could be the model's way of approximating the increased statistical probability of a critical micro-fracture existing in a longer strand. This transforms the model from an engineering tool into a scientific instrument for generating new, testable hypotheses.

### 4.1 BROADER IMPLICATIONS AND FUTURE VISION

The framework presented here is a generalizable template with broad implications for fields like additive manufacturing, biomedical engineering, and composite materials science. The vision is an automated XAI pipeline that can accelerate materials discovery by rapidly generating accurate, interpretable models from new experimental data.

### 4.2 LIMITATIONS AND AVENUES FOR FUTURE RESEARCH

The primary limitations are the dataset's scope and the lack of environmental controls. Future work should focus on creating a larger, more diverse dataset. Variations in operator reaction time and the manual application of force may have introduced minor, off-axis loads. While the model is designed to learn from this noisy data, these factors represent sources of experimental uncertainty. A promising avenue is the use of computer vision to automate measurement and, crucially, to identify and quantify surface defects as new input features for the model.

## 5 CONCLUSION

This research confronted a foundational challenge in structural mechanics. We have demonstrated that a synergistic framework combining classical theory, experimental data, and explainable machine learning can successfully bridge this gap. Our XGBoost model, informed by the principles of Euler's formula, predicted the critical buckling load of a non-ideal material with substantially lower error than a calibrated Euler baseline ($R^2 = 0.97$). The application of XAI provided a deep interpretation of the model's logic, transforming it from an opaque "black box" into a scientific instrument. This study serves as a robust proof-of-concept, illustrating that the future of modeling complex physical systems lies in the intelligent synthesis of classical theory, experimental data, and the interpretive power of modern artificial intelligence.

### ACKNOWLEDGMENTS

For the purpose of anonymous review, all acknowledgments have been omitted. The author would like to acknowledge the use of the Google Gemini LLM for polishing the manuscript and suggesting edits for clarity.

**Reproducibility Statement** All data statistics, feature definitions, model configurations, and cross-validation protocols are specified in Sections 2–4. The full dataset and source code to reproduce all experiments and figures are included in the supplementary material.

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
