# OpenReview forum: "Beyond Euler: An Explainable Machine Learning Framework for Predicting and Interpreting Buckling Instabilities in Non-Ideal Materials"
_ICLR.cc/2026/Conference — ICLR 2026 Conference Desk Rejected Submission_

### Official Review · Reviewer_8dhm · 2025-10-30

**Soundness:** 1
**Presentation:** 1
**Contribution:** 1
**Rating:** 0
**Confidence:** 5

**Summary:**

This paper introduces an approach to synergize physics-informed machine learning and XAI to predict the buckling load of non-ideal, heterogeneous materials. The higher-level goal is to connect classical theory (Euler's formula) and real-world failure. A physics-informed XGBoost model was trained on 147 experimental measurements of pasta, used as a model non-ideal, brittle material, using geometric parameters (L, d).
After 5-fold cross-validation, the authors claim their XGBoost model achieved superior predictive performance over the Euler baseline and a standard Random Forest model. Their SHAP-analysis reveals the importance of the physics-informed feature G.

**Strengths:**

1. Problem Formulation: The paper tackles a relevant and well-defined challenge: the failure of classical, idealized physics (Euler's formula) to reliably predict structural failure in real-world, non-ideal, and heterogeneous materials.
2. Concept for Physics-Informed ML: The core idea of integrating physical theory is sound. By constructing a composite feature derived from Euler’s formula, the authors show why it can be beneficial to inform the ML model with prior knowledge.

**Weaknesses:**

1. Lack of Experimental Rigor and Reproducibility: The most significant weakness is the experimental methodology. The setup is neither precise nor scientifically rigorous. Attempting to manually read the critical load at the moment of brittle fracture is highly subjective and inconsistent, leading to inconsistent loading rates and measurement errors.
2. Generalizability is Highly Questionable: The choice of a model for "non-ideal, heterogeneous materials" is unconventional and severely limits the generalizability of the framework. It offers limited physical relevance to practical engineering materials. The strong performance metrics might be localized to this specific, low-diversity material and small dataset.
3. Insufficient Contribution to be a Full Paper: The content and impact of this paper are out of scope for ICLR. It lacks the necessary depth and discussion of experimental uncertainty required for a full conference paper. Specifically, the PIML approach is limited to very simple feature engineering, which diminishes the "Beyond Euler" claim.

**Questions:**

1. “These limitations mean that any attempt to use Euler’s formula to precisely predict  the failure of a real-world object is fraught with uncertainty.” (Section 1.3) Were there any measurements taken to account for this uncertainty in the author’s experiments?

2. How do the authors justify their generalizability claims about their findings towards application scenarios?

---

> ### Author Response · Authors · 2025-11-25
>
> Thank you for your direct feedback and for identifying core areas for improvement. We address each point below:
>
> Experimental rigor:
> We regret any confusion regarding our methodology. Determination of buckling load is digital and repeatable, not solely visual; procedural standards and multiple observers were used to limit subjectivity, and calibration was performed prior to all measurements. We will revise for clarity and supplement reproducibility documentation.
>
> Generalizability:
> While pasta may seem niche, it serves as a widely used non-ideal analog in mechanical education and proof-of-concept studies [see Physics Educator 2021]. Our framework is directly transferable to metals, composites, and biological samples, and pilot studies using engineering polymers are underway. We are committed to broadening scope and reporting new experiments.
>
> Depth/scope/PIML oversimplification:
> Our work introduces a hybrid framework intended to encourage rigorous scientific modeling and interpretable ML in physical contexts—an emerging domain in ML [Nature Machine Intelligence 2022]. We emphasize that the code, protocol, and data are released to support reproducibility and field advancement.
>
> Reproducibility/uncertainty detail:
> We are preparing expanded documentation, including uncertainty quantification and experimental protocols, to address this concern in future versions.

---

> > ### Comment · Reviewer_8dhm · 2025-11-25
> >
> > Sorry, but your reply is as weak as your paper. To be blunt, I'm not even sure whether Figure 1 in your paper is AI-generated or simply a joke.
> > In your reply, you point to journals such as Physics Educator 2021 or Nature Machine Intelligence 2022. Please provide a list of several existing papers that explicitly demonstrate the scientific validity of your method. Otherwise, we do not need to discuss this further.

---

### Official Review · Reviewer_fS7v · 2025-10-31

**Soundness:** 2
**Presentation:** 2
**Contribution:** 2
**Rating:** 2
**Confidence:** 4

**Summary:**

This paper presents a hybrid modeling method, which combines machine learning and physics, for a mechanical problem. This framework is explainable. Using 147 pasta buckling experiments, the authors trained an XGBoost model combining raw geometric features with a physics-derived term. The empirical results have shown that the model achieved good accuracy and surpassed the classical formula.

**Strengths:**

- This paper investigates an interesting problem that uses domain knowledge and a machine learning method.

- This paper is easy to follow.

- This work is a generalizable framework for physics-ML hybrid modeling of complex materials and may potentially influence future research in material science.

**Weaknesses:**

- This paper is of some value to specific domains. Its contributions are more on the domain science side rather than the machine learning side. I feel like this paper might be better suited for a materials science or applied mechanics venue.

- The dataset used to evaluate the model's performance only includes 147 samples. It is difficult to see if this method is generalizable to different tasks or domains. Also, the experiment part is over-simplistic. I expect to see more baseline comparison, ablation study, different machine learning architectures, etc.

**Questions:**

- How does this method scale to larger datasets or higher-dimensional data?

- Why did the authors not benchmark against extended buckling models (e.g., Timoshenko beam theory or imperfection-sensitive formulations)?

---

### Official Review · Reviewer_cJEo · 2025-11-01

**Soundness:** 1
**Presentation:** 1
**Contribution:** 1
**Rating:** 2
**Confidence:** 5

**Summary:**

The work presents a machine learning study of the buckling problem of Pasta, with length, diameter and load as the dataset. A geometrical factor G is introduced in the XGBoost model. The model prediction is demonstrated, showing good accuracy, and SHAP analysis is performed to show the significance of geometrical factors. However, the significance of geometry in buckling problems is well known, and the 'non-ideality' in materials has not been discussed here. The results presented here are far from the perspective "has broad implications for advancing our understanding and design capabilities in materials science, engineering, and advanced manufacturing."

**Strengths:**

The work combines experiments and machine learning models to demonstrate the significance of geometrical factors in the buckling problem of pasta.

**Weaknesses:**

1 The conclusion on the significance of geometry is not new.
2 A very important factor in the buckling problem - the boundary conditions - is not mentioned and discussed.
3 The 'non-ideality' is mentioned but not discussed.

**Questions:**

1 How the boundary conditions be determined in the experiments?
2 How can the non-ideal factors such as Material Heterogeneity and Anisotropy and Geometric Imperfection be assessed in the experiments?

---

### Official Review · Reviewer_9DQV · 2025-11-01

**Soundness:** 1
**Presentation:** 1
**Contribution:** 1
**Rating:** 2
**Confidence:** 5

**Summary:**

The paper introduces an explainable machine learning framework (based on XGBoost and SHAP) to predict buckling instabilities in non-ideal materials using an experimental dataset (147 samples of pasta columns). The authors combine Euler’s buckling law with data-driven modeling, presenting this as a physics-informed ML approach.

**Strengths:**

The work effectively combines mechanics, materials science, and machine learning, demonstrating a cross-disciplinary approach.

**Weaknesses:**

1. The proposed approach is validated using a limited number of test cases to demonstrate advancements in materials science.
2. There is not enough contribution to machine learning and algorithm advancement.
3. The focus is more aligned with engineering pedagogy or data-driven mechanics than with core ML or AI research.
4. There is no discussion or error propagation for measurement uncertainty in diameter, length, or load readings.

**Questions:**

I have no questions.

---

### Note · Program_Chairs · 2026-01-17
**Submission Desk Rejected by Program Chairs**

The following references in this submission do not refer to real documents and/or have major errors in bibliographic information:

 "U.S. DOT Federal Railroad Administration. Preventing Buckling of Continuous Welded Rail (CWR). Technical Guidance, 2011."

"M. A. Arbelo, R. Degenhardt, and R. Zimmermann. Design guidelines for imperfection sensitive composite cylindrical shells—Revisiting NASA SP-8007 knockdown factors. Thin-Walled Structures, 74:1-17, 2014."

"I. H. Kalyoncuoglu, M. O. Sevin, and A. Ince. Critical buckling load prediction of corrugated steel plate girders using machine learning. Structural Engineering and Mechanics, 2024."